# Insights from a global snapshot of the change in elective colorectal practice due to the COVID-19 pandemic

**Sam E. Mason**[1]*, **Alasdair J. Scott**[1], **Sheraz R. Markar**[1], **Jonathan M. Clarke**[2],
**Guy Martin**[1], **Jasmine Winter Beatty**[1], **Viknesh Sounderajah**[1], **Seema Yalamanchili**[1],
**Max Denning**[1], **Thanjakumar Arulampalam**[3], **James M. Kinross**[1], on behalf of the
**PanSurg Collaborative**[1¶]

**1** Department of Surgery and Cancer, Imperial College London, London, United Kingdom, **2** Centre for
Mathematics of Precision Healthcare, Imperial College London, London, United Kingdom, **3** Department of
Allied Health and Medicine, Anglia Ruskin University, Cambridge, United Kingdom

¶ Collaborators in the PanSurg Collaborative are listed in the Acknowledgments.
* sam.mason@imperial.ac.uk

**Data Availability Statement:** Data are available at www.PanSurg.org.

**Funding:** The author(s) received no specific funding for this work.

## Abstract

### Background

There is a need to understand the impact of COVID-19 on colorectal cancer care globally and determine drivers of variation.

### Objective

To evaluate COVID-19 impact on colorectal cancer services globally and identify predictors for behaviour change.

### Design

An online survey of colorectal cancer service change globally in May and June 2020.

### Participants

Attending or consultant surgeons involved in the care of patients with colorectal cancer.

### Main outcome measures

Changes in the delivery of diagnostics (diagnostic endoscopy), imaging for staging, therapeutics and surgical technique in the management of colorectal cancer. Predictors of change included increased hospital bed stress, critical care bed stress, mortality and world region.

### Results

191 responses were included from surgeons in 159 centers across 46 countries, demonstrating widespread service reduction with global variation. Diagnostic endoscopy was reduced in 93% of responses, even with low hospital stress and mortality; whilst rising

**Competing interests:** The authors have declared that no competing interests exist.

critical care bed stress triggered complete cessation ($p = 0.02$). Availability of CT and MRI fell by 40–41%, with MRI significantly reduced with high hospital stress. Neoadjuvant therapy use in rectal cancer changed in 48% of responses, where centers which had ceased surgery increased its use (62 *vs* 30%, $p = 0.04$) as did those with extended delays to surgery ($p<0.001$). High hospital and critical care bed stresses were associated with surgeons forming more stomas ($p<0.04$), using more experienced operators ($p<0.003$) and decreased laparoscopy use (critical care bed stress only, $p<0.001$). Patients were also more actively prioritized for resection, with increased importance of co-morbidities and ICU need.

## Conclusions

The COVID-19 pandemic was associated with severe restrictions in the availability of colorectal cancer services on a global scale, with significant variation in behaviours which cannot be fully accounted for by hospital burden or mortality.

## Introduction

The COVID-19 pandemic has had a direct impact on global health, requiring the re-allocation and rationing of scarce resources at an unprecedented scale [1]. However, this drastic re-purposing is likely to have substantial indirect effects on the delivery of non-COVID-19 care. This is of critical importance for cancer patients, where delays to diagnosis and treatment have long term repercussions at both an individual patient and national level [2, 3]. This has been recently modelled, predicting excess deaths over the pandemic's first year of over 6000 in England and over 33,000 in the USA [4]. In the UK, it has been reported that cancer screening has been cancelled, urgent cancer referrals from the community have fallen by an average of 76% and that face-to-face outpatient appointments have been largely abandoned [4, 5]; with similar changes expected worldwide. Colorectal cancer is the third commonest cause of cancer related death globally [6], with its complex diagnostic pathways and multimodal treatment vulnerable to disruption by COVID-19 at every stage of the patient pathway. The availability of diagnostic colonoscopy and CT colonography have been severely curtailed, based on multiple international recommendations [7–9]. Similarly, resectional surgery has been cancelled in many centers due to the requirement for space (hospital beds), equipment (ventilators) as well as the redeployment of staff [2].

The speed at which the pandemic evolved was not anticipated in several countries and left little time for the generation of national or international guidance. A clearer understanding of precisely how colorectal practice has changed during the pandemic is now urgently needed, to elucidate the driving factors of change and to effectively plan for the recovery. Furthermore, many countries are anticipating a second wave of infections and services must be able to flexibly scale to maintain cancer services.

We hypothesized significant variation in the organisational responses of colorectal centers globally during the COVID-19 pandemic. In this snapshot survey of colorectal practice, we collected survey data to determine the local impact on colorectal cancer diagnostic and therapeutic domains.

## Materials and methods

This study was a global cross-sectional survey of the change in elective colorectal practice caused by the COVID-19 pandemic.

## Survey design and data collection

A survey was designed through consultation of a core committee of surgeons experienced in qualitative research methodology. Questions were a combination of multiple choice and forced-ranking scales (S1 File). The questions were designed to cover demographic information of each respondent and center, with exploration of how diagnostic and therapeutic resources, treatment strategies and personnel allocation had been affected by the pandemic. This survey was administered using Google Forms (Google LLC, USA) and invites were distributed to international colleagues, through governing bodies such as the Royal College of Surgeons of England and through the PanSurg social medial channels; from 2nd April 2020. Consultant or attending surgeons performing elective resections for colorectal cancer were eligible for inclusion. National mortality was defined as the mortality from COVID-19 for each country on the day the survey was received [10], reported as deaths/million.

## Definitions of metrics used

COVID-19 Load was defined as the number of COVID-19 patients currently admitted to the respondent's center and was categorised as low (0–20), medium (21–100) and high (>100) patients). Total hospital bed capacity was stratified as low (0–500), medium (501–1000) and high (>1000 beds); with critical care capacity also described as low (0–20), medium (21–100) and high (>100 beds). By comparing the COVID-19 Load to the hospital and critical care bed capacity in of each respondents' center, we derived the metrics *Hospital Bed Stress* and *Critical Care Bed Stress*. Stress was low when the strata of capacity exceeded the COVID-19 Load, medium when the strata of COVID-19 Load equaled that of bed capacity and high when the strata of COVID-19 Load exceeded that of capacity. For large hospitals where the COVID burden was also high, stress was determined to be high if the total patient burden was greater than 200.

Surgeons were asked to rank six factors for importance when prioritizing a patient for colorectal resection. A pairwise comparison was made of the ordinal perceived importance between each factor within a response and then aggregated across the dataset. The *Priority Score* was defined for each factor as the sum of the scores against the other 5 factors, where a higher score reflects the surgeon valuing it with greater general importance and the converse true of negative scores. The absolute value relates to the strength of preference.

To interpret deviation from usual practice across 11 domains (endoscopy, computed tomography (CT), Magnetic Resonance Imaging (MRI), Positron Emission Tomography (PET), therapeutic endoscopy, rectal neoadjuvant therapy, colon neoadjuvant therapy, delays to surgery, operators, laparoscopy, stoma formation) we defined *a Change* Score. For each domain 1 point was given if a respondent's center had limited availability and 2 were given if it was unavailable.

## Statistical analysis

Data were processed and analysed in R Studio v1.1.453 using the 'tidyverse' package [11, 12]. Categorical variables were analysed using Chi-squared or Fisher's Exact test based on expected cell values. Continuous data were analysed using ANOVA or Mann Whitney U test for parametric and non-parametric data respectively. Statistical significance was defined using $\alpha$ = 0.05.

## Results

198 survey responses were received until 17th May 2020, with 7 excluded from analysis due to insufficient data (n = 6) or respondent not being a consultant or attending surgeon (n = 1).

This left 191 replies for inclusion, from 159 distinct centers across 46 countries. The demographics and characteristics of the included surgeons and centers are presented in Table 1.

## COVID-19 burden and stress on hospital resources

162 of the respondents (85%) described the presence of patients suffering with COVID-19 in their center, with a relative balance across low, medium and high burdens. The matrices comparing COVID Load with hospital and critical care bed capacities are demonstrated in Fig 1, defining the metrics *Hospital Bed Stress* and *Critical Care Bed Stress*. These metrics were validated by comparing the strata to national mortality, demonstrating significant association ($p = <0.001$, Fig 2). Global variation in COVID Load, *Hospital Bed Stress* and *Critical Care Bed Stress* is presented in Table 2.

## Impact of system stress on diagnosis and therapeutics

COVID-19 has had a dramatic impact on the capability of centers to provide diagnostic and therapeutic services for colorectal cancer (Table 3), with significant variation in the extent of change across available modalities and how this correlates to the stress of the healthcare center.

**Table 1. Characteristics of the surgeons and centers responding to the survey.**

| Characteristic | | n (%) |
|---|---|---|
| Gender | Male | 164 (86) |
| | Female | 27 (14) |
| Specialty | Colorectal | 159 (83) |
| | General | 32 (17) |
| Hospital Setting | Tertiary/Academic | 151 (79) |
| | Local/District | 40 (21) |
| World Region | Europe | 105 (55) |
| | Australasia | 40 (21) |
| | Asia | 22 (12) |
| | The Americas | 20 (10) |
| | Africa | 3 (2) |
| | Middle East | 1 (1) |
| COVID-19 Patient Load | 0 | 29 (15) |
| | 1–9 | 44 (23) |
| | 10–20 | 18 (9) |
| | 21–50 | 27 (14) |
| | 51–100 | 31 (16) |
| | 101–200 | 23 (12) |
| | >200 | 19 (10) |
| Hospital Bed Capacity | <200 | 14 |
| | 201–500 | 70 |
| | 501–1000 | 73 |
| | >1000 | 34 |
| Critical Care Bed Capacity | <21 | 89 |
| | 21–50 | 64 |
| | 51–100 | 27 |
| | >100 | 11 |

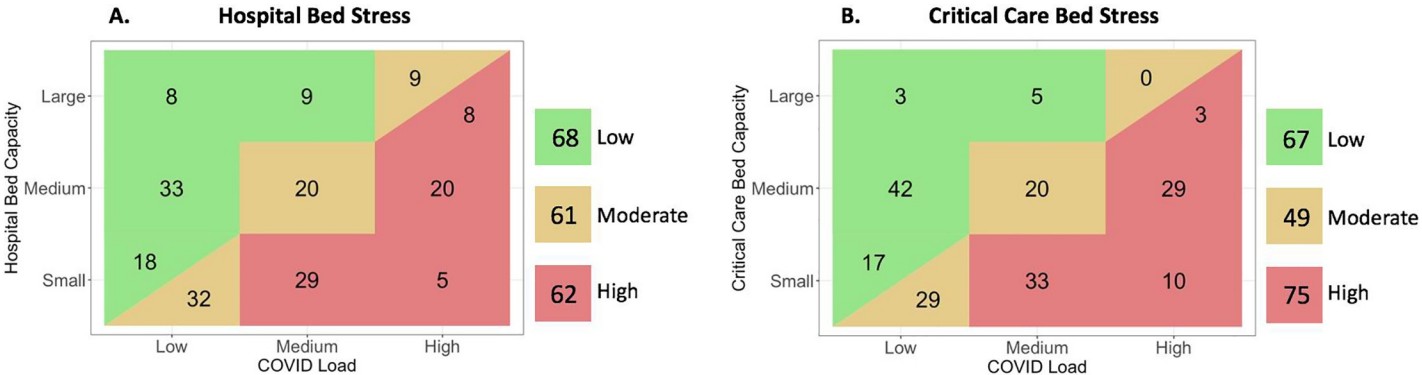

**Fig 1.** Low, moderate and high *Hospital Bed Stress* (A) and *Critical Care Bed Stress* (B), derived from the *COVID-19 Load* compared to the hospital and critical care bed capacities respectively. Centers with low hospital or critical care bed capacity and no COVID patients were determined to have a low stress, whereas similarly sized centers caring for <20 patients deemed at moderate stress. Centers with high hospital or critical care bed capacity and greater than 200 COVID patients were determined to have a high stress, whereas similarly sized centers caring for <200 patients deemed at moderate stress.

This is most notable when examining the availability of diagnostic endoscopy, which is operating at normal levels in only 7% of responses. Having no diagnostic endoscopy rather than a limited one was associated with high *Critical Care Bed Stress* ($p = 0.02$) and national mortality ($p < 0.001$), however, not with *Hospital Bed Stress* ($p = 0.24$). There are slight differences for therapeutic endoscopy. Of the 176 respondents that used it prior to the pandemic, there was greater success in maintaining a normal service (23%), however, there is again evidence that many centers were limiting it despite low stress. The hospital and critical care bed stress metrics appear unable to account for why some units were ceasing this service entirely. CT, MRI and PET showed a 40–49% reduction, with the impact on both MRI and PET scanning appearing to be correlated with *Critical Care Bed Stress* and *Hospital Bed Stress*.

The COVID-19 pandemic has prompted significant changes in the use of neoadjuvant oncologic treatments, with 48% of centers applying these differently for rectal and 23% for colon cancer. If practice has changed, it is more likely to be an increase in use for rectal cancer

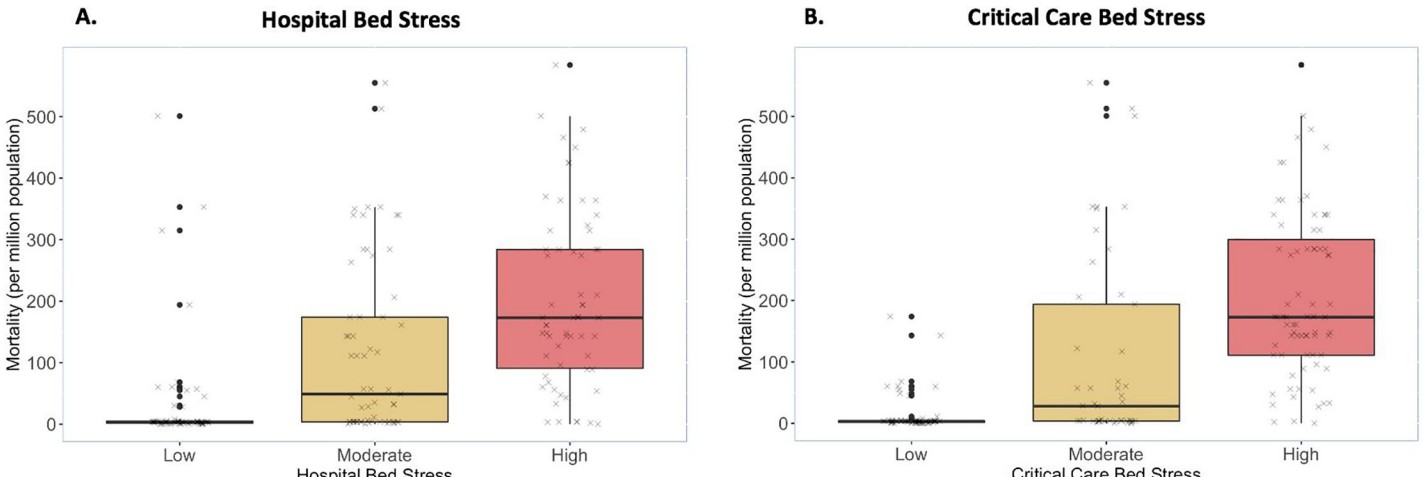

**Fig 2.** Box and whiskers plot of the relationship between national mortality rate from COVID-19 and either *Hospital Bed Stress* (A) or *Critical Care Bed* Stress (B). Black circles represent outliers and 'x' is a jitter plot of the raw data. $p = <0.001$ for all comparisons between groups.

**Table 2. The global variation of COVID Load, *Hospital Bed Stress* and *Critical Care Bed Stress*.**

| World Region | n | COVID Load (%) | | | Hospital Bed Stress (%) | | | Critical Care Bed Stress (%) | | |
|---|---|---|---|---|---|---|---|---|---|---|
| | | Low | Medium | High | Low | Moderate | High | Low | Moderate | High |
| Europe | 105 | 24 | 39 | 37 | 17 | 33 | 50 | 11 | 22 | 67 |
| Australasia | 40 | 95 | 5 | - | 60 | 38 | 3 | 75 | 25 | - |
| Asia | 22 | 86 | 9 | 5 | 86 | 5 | 9 | 73 | 18 | 9 |
| The Americas | 20 | 30 | 60 | 10 | 30 | 40 | 40 | 35 | 50 | 15 |
| Other | 4 | 75 | 25 | - | 25 | 50 | 25 | 50 | 50 | - |

Note: Some percentage sums may not equal 100 due to rounding.

with no clear preference in colon cancer. Which change is made does not correlate with the country or hospital stress. There appears to be specific associations with an increased use of neoadjuvant therapies rather than a decrease or no change. Centers that have stopped resectional surgery entirely are significantly more likely to increase use of neoadjuvant therapy in rectal cancer (62 *vs* 30%, $p = 0.04$) but not colon cancer (31 *vs* 11%, $p = 0.11$). In centers where resectional surgery is continuing, greater use of neoadjuvant therapies is associated with expecting an extended delay to surgery for rectal cancers ($p = 0.001$).

The impact of COVID-19 on the operative approach, strategy and personnel for elective colorectal cancer is marked. 7% of centers have ceased elective resections entirely. Of those continuing to operate, there is an increased use of attending surgeons as the principle and assisting surgeon, strongly associated with high *Hospital Bed Stress* and *Critical Care Bed Stress*. There is no standardised approach for preoperative patient screening with centers using PCR (33%), imaging (most commonly CT chest, 15%), both PCR and imaging (17%), risk survey (4%), clinical assessment (3%), both clinical and risk survey (2%) and a combination of serology, imaging and PCR (2%). No screening is performed in 24% of responses and there is no clear association with geographical region. There has been a dramatic reduction in the use of laparoscopy (48%), with reducing or ceasing laparoscopy strongly associated with *Critical Care Bed Stress* ($p = <0.001$), but not to country, national mortality ($p = 0.09$) or Hospital Bed Stress ($p = 0.14$). Of those using laparoscopy, 77% are deploying smoke extraction devices. 54% of centers have increased their stoma formation rate, largely within the context of left-sided resections (more likely to perform a Hartmann's procedure or to defunction a primary anastomosis). Increased stoma formation was significantly associated with increased national mortality ($p = 0.02$) and both metrics of hospital stress ($p = <0.04$).

## Change Score

The *Change Score* across 11 domains (endoscopy, CT, MRI, PET, therapeutic endoscopy, rectal neoadjuvant therapy, colon neoadjuvant therapy, delays to surgery, operators, laparoscopy, stoma formation) demonstrates that on average, 7 aspects of patient care have changed, with degree of change significantly increasing as *Hospital Bed Stress* and *Critical Care Bed Stress* increase ($p = 0.007$ and $<0.001$ respectively, Fig 3).

## Prioritisation of patients when scheduling for theatre

Of the 93% of responses where elective operating had not ceased, 64% were implementing new strategies when prioritizing patients for resection, with guidance generated at a local (62%) or national level (38%). The *Priority Score* for the six ranked variables could be calculated across

**Table 3. Change in diagnostic and therapeutic capabilities stratified by *Hospital Bed Stress* and *Critical Care Bed Stress*.**

| | | *Hospital Bed Stress* | | | | *Critical Care Bed Stress* | | | |
|---|---|---|---|---|---|---|---|---|---|
| | Availability | Low | Moderate | High | *p* value | Low | Moderate | High | *p* value |
| **Diagnostic Endoscopy** | Normal | 6 | 4 | 3 | 0.24 | 4 | 6 | 3 | **0.02** |
| | Limited | 55 | 43 | 44 | | 56 | 36 | 50 | |
| | Unavailable | 7 | 14 | 15 | | 7 | 7 | 22 | |
| **CT** | Normal | 46 | 35 | 33 | 0.222 | 45 | 31 | 38 | 0.11 |
| | Limited | 22 | 26 | 29 | | 22 | 18 | 37 | |
| **MRI** | Normal | 44 | 39 | 29 | **0.03** [a] | 45 | 30 | 37 | **0.049** [a] |
| | Limited | 21 | 20 | 33 | | 19 | 18 | 37 | |
| | Unavailable | 3 | 2 | - | | 3 | 1 | 1 | |
| **PET** | Normal | 39 | 40 | 19 | **0.001** | 41 | 28 | 29 | **0.04** |
| | Limited | 20 | 16 | 35 | | 17 | 17 | 37 | |
| | Unavailable | 9 | 5 | 8 | | 9 | 4 | 9 | |
| **Therapeutic Endoscopy**[b] | Normal | 17 | 15 | 9 | 0.10 | 19 | 8 | 14 | 0.34 |
| | Limited | 33 | 36 | 30 | | 32 | 28 | 39 | |
| | Unavailable | 10 | 8 | 18 | | 10 | 8 | 18 | |
| **Rectal Neoadjuvant Rx** | Normal | 40 | 31 | 28 | 0.19 | 41 | 26 | 32 | 0.12 |
| | Increased | 17 | 18 | 26 | | 14 | 16 | 31 | |
| | Decreased | 11 | 13 | 7 | | 12 | 7 | 12 | |
| **Colon Neoadjuvant Rx** | Normal | 50 | 49 | 49 | 0.79 | 53 | 38 | 57 | 0.65 |
| | Increased | 11 | 7 | 6 | | 8 | 8 | 8 | |
| | Decreased | 7 | 5 | 7 | | 6 | 3 | 10 | |
| **Delay After Neoadjuvant Rx (weeks)** | No delay | 25 | 27 | 14 | **0.01** | 29 | 17 | 20 | **3** |
| | <2 | 12 | 6 | 6 | | 8 | 10 | 6 | |
| | 2–4 | 16 | 8 | 10 | | 15 | 5 | 14 | |
| | 4–8 | 10 | 7 | 17 | | 6 | 9 | 19 | |
| | >8 | 5 | 13 | 15 | | 9 | 8 | 16 | |
| **Operator** | Dual consultants | 13 | 25 | 20 | **0.006** | 11 | 17 | 30 | **0.005** |
| | Consultant, trainee assisted | 44 | 31 | 28 | | 45 | 26 | 32 | |
| | Trainee under supervision | 9 | 3 | 5 | | 9 | 4 | 4 | |
| | NA—no resections | 2 | 2 | 9 | | 2 | 2 | 9 | |
| **Use of Laparoscopy** | Normal | 41 | 31 | 26 | 0.14[c] | 41 | 29 | 28 | **< 0.001**[c] |
| | Decreased | 13 | 18 | 24 | | 2 | 13 | 30 | |
| | Ceased | 14 | 13 | 10 | | 13 | 7 | 17 | |
| | Increased | - | 1 | - | | 1 | - | - | |
| **Use of Stomas** | Normal | 35 | 28 | 18 | **0.03**[d] | 35 | 26 | 20 | **0.002**[d] |
| | Decreased | 3 | 1 | 1 | | 3 | 1 | 1 | |
| | Increased | 29 | 32 | 40 | | 28 | 22 | 51 | |

CT–Computed Tomography; MRI–Magnetic Resonance Imaging; PET–Positron Emission Tomography; Rx–treatment.

[a]'Limited' *vs* 'Normal';

[b]only included when therapeutic endoscopy available prior to COVID-19

[c]'Normal' *vs* 'Decreased' *vs* 'Ceased';

[d]'Normal' *vs* 'Increased'.

the *Hospital Bed Stress* and *Critical Care Bed Stress* strata (Table 4). When plotted, the relative linearity of the line denoting all responses demonstrates clear transitive ranks of the variables based on priority (Fig 4); with the ranks in descending order of importance:

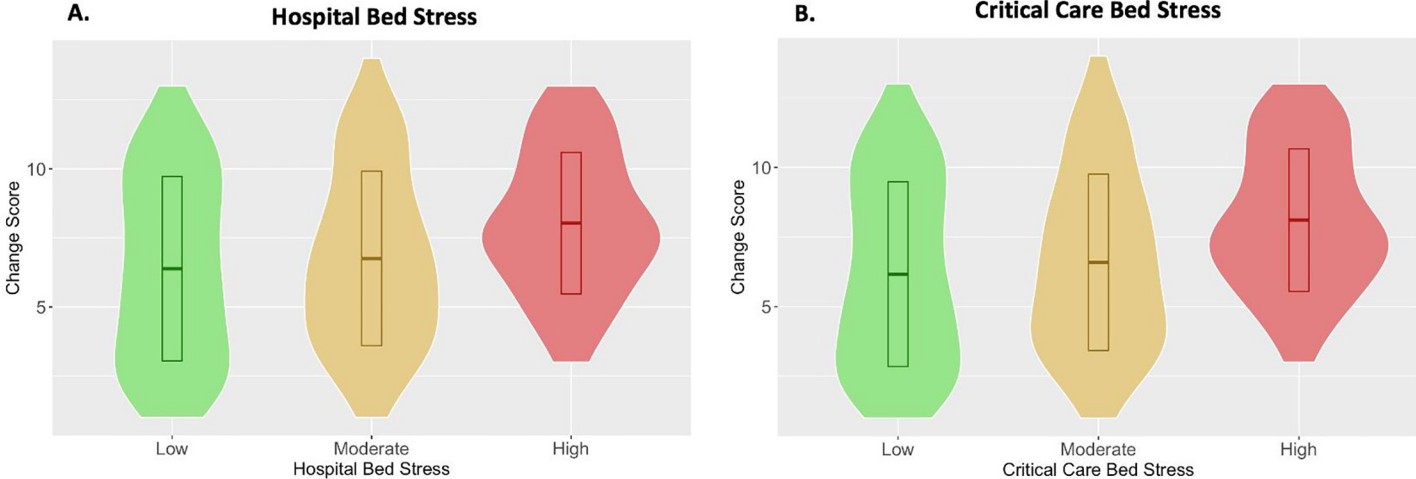

**Fig 3. Violin plots of *Change Score* against *Hospital Bed Stress* and *Critical Care Bed Stress* metrics.** The summary boxes denote the mean +/- standard deviation for each group. The groups are statistically significantly different ($p$ = 0.007 and <0.001 respectively).

1. Co-morbidities

2. Disease stage

3. Need for ICU bed postoperatively

4. Extended delay following neoadjuvant therapy

5. Age

6. Expected difficulty of case

Surgeons were prioritizing patients for surgery differently based on the *Hospital Bed Stress*. As it increases, co-morbidities became relatively more important, with extended delay following neoadjuvant therapy and expected case difficulty becoming less important (Fig 4A). This was such that for High *Hospital Bed Stress* and *Critical Care Bed Stress*, age surpassed extended delay in priority; findings unique to this stratum. The findings for *Critical Care Bed Stress* were generally similar to *Hospital Bed Stress*, however, as burden increases there is slightly greater relative importance placed on the need for an ICU bed postoperatively (Fig 4B). The difference

**Table 4. *Priority Scores* for the six variables considered when scheduling patients for theatre, presented for all responses and by Hospital Bed Stress and Critical Bed Stress strata.**

| Variable | All | Hospital Bed Stress | | | Critical Care Bed Stress | | |
|---|---|---|---|---|---|---|---|
| | | High | Moderate | Low | High | Moderate | Low |
| Co-morbidity | 337 | 397 | 426 | 191 | 438 | 238 | 285 |
| Disease Stage | 272 | 206 | 348 | 222 | 155 | 378 | 282 |
| Need for ICU Bed | 98 | 166 | -3 | 104 | 150 | 78 | 31 |
| Extended Delay After Neoadjuvant Therapy | -108 | -185 | -150 | 11 | -188 | -47 | -51 |
| Age | -183 | -151 | -178 | -177 | -102 | -265 | -174 |
| Expected Case Difficulty | -416 | -434 | -442 | -351 | -453 | -382 | -373 |

A higher score represents a greater importance of the variable, with the score scaled by total responses within each stress strata.

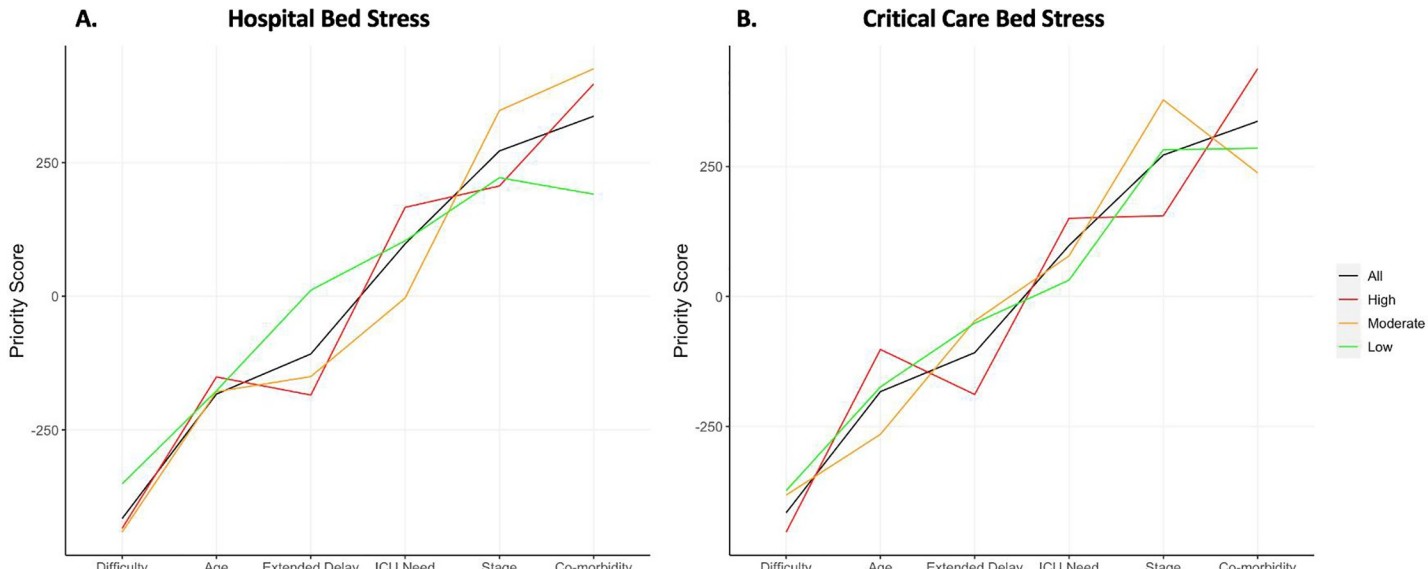

**Fig 4.** Line plot of the *Priority Scores* for each variable when scheduling a patient for theatre, with the impact of high, moderate and low *Hospital Bed Stress* (A) and *Critical Care Bed* Stress (B) compared. Note: Higher score demonstrates a higher priority.

between the highest and lowest priority scores increases with stress, demonstrating that when under pressure, surgeons place a higher importance on these variables.

## Discussion

This is a cross-sectional survey of the global change in elective colorectal cancer practice as a consequence of the COVID-19 pandemic, evaluating 191 responses from 46 countries. The primary finding was that whilst practice change has been widespread, there is considerable variation in the response between centers. This can sometimes be understood within the context of system burden (*Hospital Bed Stress*, *Critical Care Bed Stress*), national burden (mortality); however, in many cases triggers for change cannot be directly identified.

The vast majority of centers globally have had to limit the care they provide across diagnostic endoscopy, imaging for staging, neoadjuvant treatments and definitive surgery. The *Critical Care Bed Stress* metric appears to be one of the best predictors of change, likely due to these facilities experiencing the greatest demand during the pandemic. This would prompt the reallocation of resources from other services in the hospital—endoscopy units, theatres, and recovery areas are all likely to be designated as contingency zones for additional capacity; with doctors and nurses of other specialties redeployed here. However, there were many examples of centers limiting their cancer care despite having low stress and mortality. Diagnostic endoscopy is an excellent example of this. It appears that the initial response to COVID-19 was to limit services even when hospital pressure is low, with rising *Critical Care Bed Stress* or national mortality the triggers to cease services entirely. Concern over diagnostic endoscopy being an aerosol generating procedure was unlikely to be solely responsible for low-stress centers limiting their service, as these centers were no more likely to reduce laparoscopy use or to deploy filtered smoke extractors. Conversely, there were centers experiencing high stresses who were able to provide care without excessive limitation. Understanding and predicting this variance is challenging and given that it does not seem correlated to the country (and therefore the guidance from relevant governing bodies), it is possible that there are local coping mechanisms employed at certain centers which will need to be identified in future research.

Guidance issued from regulatory bodies frequently lacked the clarity demanded by clinicians, could become outdated in weeks and often conflicted with the guidance of other bodies. Considering the disparity in COVID-19 impact worldwide, future guidelines will require marked flexibility to allow effective application in individual centers.

Behaviour changes in surgical strategy and technique demonstrate high correlation with increased *Hospital Bed Stress* and *Critical Care Bed Stress*, with changes designed to reduce risk to both patients and operating room staff. Whilst there has been no clear evidence that laparoscopy causes aerosolisation of the SARS-CoV-2 virus, this is the case with other viruses and SARS-CoV-2 has been found in peritoneal fluid [13, 14]. This concern has caused conflicting advice from regulatory bodies, with an intercollegiate paper including the Royal College of Surgeons recommending a general stop to laparoscopy with other bodies recommending laparoscopy continuing with additional safeguards such as smoke filtration [15–17]. Surgeons have taken the cautious approach by limiting or ceasing laparoscopy by almost half, particularly with higher mortality; but there is still inadequate appetite for risk mitigation given only 77% use smoke filtration. Preoperative COVID-19 screening is universally recommended but still not standard of care in many centers [18–21]. The use of stomas was not driven by international guidance and appears subject to individual surgeon preference. Defunctioning left-sided anastomoses and/or having a greater propensity to perform a Hartmann's procedure is likely to decrease the initial use of hospital resources and protect patient safety in the short-term (by avoiding the risk and severity of anastomotic leakage). As services are restarted, stoma reversal may struggle to compete for theatre space increasing the risk of stoma formation being permanent.

The variation in the application of neoadjuvant therapies for colorectal cancers is marked. The fact that whether it an increase or decrease was made did not correlate with the country, mortality or hospital stress implies it is driven by local practice from clinicians present at the multidisciplinary meetings. The concern with such a cause of variation is that it is less likely to be evidence-based and the lack of consensus suggests wider guidance is not being applied. The greater propensity to change in rectal rather than colon cancer may be that the evidence base is stronger in this cohort and centers have more experience in its use. There was a strong indication that in centers with extended delays to surgery or without resection happening, neoadjuvant therapies were increasingly used. This provides oncologic therapy to slow progression or cause down-staging [22]. The use of chemoradiation in rectal cancer can cause complete clinical response in 7–27% of cases [23], with neoadjuvant chemotherapy in colon cancer appearing to improve oncological outcomes [24]. The concern is that the patients upon which these studies focus had advanced cancers and it is unclear if the same benefits apply to earlier stages. Furthermore, unlike more aggressive cancers such as of the esophagus and pancreas; the relationship between treatment delay and disease progression is much less clear in colorectal, with studies offering different perspectives between harmful delays in addition to the difference between colon and rectal cancers [25–27]. The optimal time between neoadjuvant therapy and operative intervention is also unclear, with some evidence that delay beyond 6–8 weeks could be beneficial, however, the point at which extended delay becomes harmful has not been defined [28]. Whilst hospital visits for chemoradiation and the resultant immunosuppression increase a patient's risk from COVID-19, the balance between oncologic benefit and COVID-19 risk is not clear and evidence is currently lacking. Centers should be flexible and dynamic as new guidance and evidence becomes available [29]. Data on long-term oncological outcomes following COVID-19 is eagerly awaited and will be necessary to guide strategies if this pandemic is prolonged or when a similar threat arises in the future.

As the burden from COVID-19 increases, surgeons change their approach to prioritising patients for theatre, likely as a necessity to identify which patients would benefit most when

access is limited. Co-morbidities appear to be of the highest importance. High *Critical Care Bed Stress* was associated with increased importance of needing an ICU bed postoperatively with age increasingly considered given the disproportionate mortality as age increases. Risk prediction tools such as the American College of Surgeons calculator would better support decision-making if the impact of COVID-19 was incorporated [30].

Although the key findings of this study are variance in behaviours and practice due to the COVID-19 pandemic, it may be that these could be better predicted by factors not collected by the survey, for example, patient choice regarding treatment strategies. There is a risk of selection biases caused by the survey being in English and that the distribution was likely to favour Europe and high-income countries. It was not possible to make extensive comparisons by country or region given that COVID-19 has disproportionally impacted different world areas and therefore variation is not sufficient for meaningful comparisons on the smaller scale. Whilst this may not reveal nuanced behaviours within world regions, by taking a global perspective, this work has been successful in making predictions of variation whilst considering confounding factors such as national mortality. The self-reporting of outcomes risks response bias, where response inaccuracies may be caused by individual personality, psychology or data availability. The findings here do compare current practice to baseline however the degree of change is cross-sectional and therefore trends over time cannot be explored. To address this, the survey will continue to collect responses at: http://tiny.cc/4tkbpz.

This work has implications for research and clinical practice, both within colorectal surgery and for wider service management during emergency scenarios. In future versions of the survey, questions will need to assess the evolving use of healthcare networks, either between public-public or public-private partnerships; where centers have been increasingly cooperating to support each other and provide contingency capacity. If there is a second pandemic wave, this project has highlighted that it is possible to protect many vital elective colorectal services even when the COVID burden is high, which will be crucial to reducing non-COVID deaths. The change in oncologic practice which was identified, particularly the different use of neoadjuvant therapies, provides a unique opportunity to study cancer outcomes with these different treatment strategies. Studies will be required to determine if these new approaches may have improved patient outcome (whether oncologic or quality of life), informing whether prospective interventional trials should be conducted.

## Conclusion

This study gives new insight into elective colorectal cancer practice on a global scale, with wide geographical coverage and a range of COVID-19 burdens. Whilst there is evidence of widespread limitation of services, there is significant variation in behaviours which hospital burden and national mortality cannot fully account for. The colorectal cancer clinical community requires best practice to be defined based on consensus and emerging evidence, with improved information transfer and learning between centers; especially from those better coping with this pandemic.

## Supporting information

**S1 File.**
(PDF)

## Acknowledgments

The authors would like to acknowledge the contribution of all in the PanSurg Collaborative: Sanjay Purkayastha, Nikita Rathod, Rabiya Aseem, Emily Deurloo, Ayush Kulshreshtha,

Maxwell Flitton, Roger van Schie, Liam Poynter, Ravi Aggarwal, Simon Dryden, Ola Markiewicz, Piers Boshier, Leigh Warren, Osama Moussa, Ee Teng Goh, Melanie Almonte, Swathikan Chadminbaran, Jan Przybylowicz, Simon Erridge, Uddhav Vaghela, Simon Rabinowicz, Paris Bratsos, Ovidiu Serban.

## Author Contributions

**Conceptualization:** Sam E. Mason, Alasdair J. Scott, Sheraz R. Markar, Jonathan M. Clarke, Guy Martin, Jasmine Winter Beatty, Viknesh Sounderajah, Seema Yalamanchili, Max Denning, James M. Kinross.

**Data curation:** Sam E. Mason, Alasdair J. Scott, Jonathan M. Clarke, Jasmine Winter Beatty, Viknesh Sounderajah.

**Formal analysis:** Sam E. Mason, Alasdair J. Scott, Sheraz R. Markar, Jonathan M. Clarke.

**Methodology:** Sam E. Mason, Alasdair J. Scott, Sheraz R. Markar, Jonathan M. Clarke, Guy Martin, Viknesh Sounderajah, Seema Yalamanchili, Max Denning, Thanjakumar Arulampalam.

**Project administration:** Jasmine Winter Beatty, Seema Yalamanchili.

**Resources:** Max Denning.

**Software:** Alasdair J. Scott.

**Supervision:** Thanjakumar Arulampalam, James M. Kinross.

**Writing – original draft:** Sam E. Mason, Guy Martin, James M. Kinross.

**Writing – review & editing:** Sam E. Mason, Alasdair J. Scott, Sheraz R. Markar, Jonathan M. Clarke, Guy Martin, Jasmine Winter Beatty, Viknesh Sounderajah, Seema Yalamanchili, Max Denning, Thanjakumar Arulampalam, James M. Kinross.

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
