## [Decision Letter · Decision Letter 0]

18 Aug 2020

PONE-D-20-17999

Insights from a global snapshot of the change in elective colorectal practice due to the COVID-19 pandemic

PLOS ONE

Dear Dr. Mason,

Thank you for submitting your manuscript to PLOS ONE. After careful consideration, we feel that it has merit but does not fully meet PLOS ONE’s publication criteria as it currently stands. Therefore, we invite you to submit a revised version of the manuscript that addresses the points raised during the review process.

We look forward to receiving your revised manuscript.

Kind regards,

Zhi Ven Fong, M.D., M.P.H.

Academic Editor

PLOS ONE

Journal Requirements:

3.One of the noted authors is a group or consortium [insert name of group or team]. In addition to naming the author group, please list the individual authors and affiliations within this group in the acknowledgments section of your manuscript. Please also indicate clearly a lead author for this group along with a contact email address.

Reviewers' comments:

Reviewer's Responses to Questions

**Comments to the Author**

1. Is the manuscript technically sound, and do the data support the conclusions?

Reviewer #1: Yes

Reviewer #2: Yes

2. Has the statistical analysis been performed appropriately and rigorously? 

Reviewer #1: Yes

Reviewer #2: Yes

3. Have the authors made all data underlying the findings in their manuscript fully available?

Reviewer #1: Yes

Reviewer #2: Yes

4. Is the manuscript presented in an intelligible fashion and written in standard English?

Reviewer #1: Yes

Reviewer #2: Yes

5. Review Comments to the Author

Reviewer #1: This is a well-written paper that uses survey data from colorectal surgeons around the globe to highlight the important impact that COVID-19 has had on care of colorectal cancer patients. While many of the conclusions are expected, they do well to summarize important alterations in practice using their survey data

Intro, pg 2 line 96: Would remove “prospectively” and just say “collected survey data”. Prospectively implies that one would follow a cohort or outcome measure over time after the initial survey

Methods: Were surgeons specifically sent invitations or were surgeons directed to a study info page/registration page on the society websites/social media pages, or were invites sent to all members of the societies? If exact number of invites is known, it should be presented. In regard to the critical care bed stress and hospital bed stress, you define high load as a strata of COVID-19 exceeding capacity, however one could argue that a hospital with high hospital bed and critical bed size with high covid-10 load is under significant stress – however by your definition, it would be considered medium stress. It is impossible for hospitals with high bed/ICU capacity to be in the high bed stress loads based on this definition, correct? Perhaps representing bed stress on % of beds filled with COVID-19 patients? Table 1 should also include breakdown of hospitals by number of hospital beds and ICU beds.

Results: There are 191 respondents from 159 centers. This implies that some centers have more than one surgeon replying. Are these double counted in the results?

Otherwise the results are well presented

Reviewer #2: This is an interesting article that aims to explore the impact of the COVID 19 pandemic on the diagnosis and treatment of colorectal cancer. The authors conducted a global survey to assess the response of colorectal and general surgeons around the world.

Major concern: There is significant bias since the study has included a large number of countries with striking differences in healthcare systems, systematic response to the pandemic, a wide variation in COVID incidence/mortality, etc. With this degree of bias it is really difficult to make comparisons and come up with sound conclusions.

There are some interesting questions asked in the survey, but we do not get the full picture based on the answers provided. For instance, does the dramatic decrease in use of laparoscopy mean that there was an equivalent increase in open surgery or that the operations were postponed to a later date?

Would like to see additional tables from some of the analyses that the authors describe in the last 2 paragraphs of the results.

What are the future steps that the authors suggest based on the results of the study?

6. PLOS authors have the option to publish the peer review history of their article (what does this mean?). If published, this will include your full peer review and any attached files.

Reviewer #1: No

Reviewer #2: No

---

## [Author Response · Author response to Decision Letter 0]

3 Sep 2020

Reviewer 1:

1. Would remove “prospectively” and just say “collected survey data”. Prospectively implies that one would follow a cohort or outcome measure over time after the initial survey. This change has been made. 

2. Were surgeons specifically sent invitations or were surgeons directed to a study info page/registration page on the society websites/social media pages, or were invites sent to all members of the societies? If exact number of invites is known, it should be presented. In some cases invitations were sent directly to individuals however the survey link was also published on our website and through Twitter. As a result, it is not possible to give a reliable estimate of the total number of surgeons who received an invitation. 

3. In regard to the critical care bed stress and hospital bed stress, you define high load as a strata of COVID-19 exceeding capacity, however one could argue that a hospital with high hospital bed and critical bed size with high covid-10 load is under significant stress – however by your definition, it would be considered medium stress. It is impossible for hospitals with high bed/ICU capacity to be in the high bed stress loads based on this definition, correct? Perhaps representing bed stress on % of beds filled with COVID-19 patients? Thank you for highlighting this excellent point. It is correct that there were a small number of large hospitals (>1000 beds) who had high COVID burdens and were described as having a medium overall stress. This group has been split into 2, based on whether <200 (medium stress) or >200 (high stress) total COVID patients were being treated. The plots, tables and text have been updated to reflect this; with no dramatic changes to the results. Unfortunately, % beds filled cannot be used as a metric as data was collected as ranges in categories (eg. 20-50 patients, 50-100 patients etc). 

4. Table 1 should also include breakdown of hospitals by number of hospital beds and ICU beds. Table 1 has been updated with this data. 

5. There are 191 respondents from 159 centers. This implies that some centers have more than one surgeon replying. Are these double counted in the results? Unfortunately, we were unable to determine the centre for some of the responses but were able to identify 159 unique ones. There were approximately 10 duplicates (10 centres with 2 replies each), but we found that responses were spread throughout the collection period and therefore, it was deemed valuable data as it gives information at different time points. 

Reviewer 2:

1. There is significant bias since the study has included a large number of countries with striking differences in healthcare systems, systematic response to the pandemic, a wide variation in COVID incidence/mortality, etc. With this degree of bias it is really difficult to make comparisons and come up with sound conclusions. We believe that the variation described here is the strength of the work and it was necessary to sample such variation in order to address the research questions of the project. Eg. When determining causes for centres to limit services, it is vital to make comparisons between regions with high and low COVID burden, different sizes hospitals, different national mortalities etc. 

2. There are some interesting questions asked in the survey, but we do not get the full picture based on the answers provided. For instance, does the dramatic decrease in use of laparoscopy mean that there was an equivalent increase in open surgery or that the operations were postponed to a later date? The survey only collected data on whether the centre was still operating electively and if so, whether laparoscopy was used differently. We appreciate that another interesting question is whether the overall volume has changed, however that was not one of the data points collected. 

3. Would like to see additional tables from some of the analyses that the authors describe in the last 2 paragraphs of the results. A table of the raw data across all groups has been inserted (table 4). 

4. What are the future steps that the authors suggest based on the results of the study? A paragraph has been added to the discussion section, documenting the implications of the study for future research and pandemic management in the case of a second wave.

---

## [Decision Letter · Decision Letter 1]

28 Sep 2020

Insights from a global snapshot of the change in elective colorectal practice due to the COVID-19 pandemic

PONE-D-20-17999R1

Dear Dr. Mason,

We’re pleased to inform you that your manuscript has been judged scientifically suitable for publication and will be formally accepted for publication once it meets all outstanding technical requirements.

Kind regards,

Zhi Ven Fong, M.D., M.P.H.

Academic Editor

PLOS ONE

Additional Editor Comments (optional):

Reviewers' comments:

Reviewer's Responses to Questions

**Comments to the Author**

1. If the authors have adequately addressed your comments raised in a previous round of review and you feel that this manuscript is now acceptable for publication, you may indicate that here to bypass the “Comments to the Author” section, enter your conflict of interest statement in the “Confidential to Editor” section, and submit your "Accept" recommendation.

Reviewer #1: All comments have been addressed

Reviewer #2: All comments have been addressed

2. Is the manuscript technically sound, and do the data support the conclusions?

Reviewer #1: Yes

Reviewer #2: Yes

3. Has the statistical analysis been performed appropriately and rigorously? 

Reviewer #1: Yes

Reviewer #2: Yes

4. Have the authors made all data underlying the findings in their manuscript fully available?

Reviewer #1: Yes

Reviewer #2: Yes

5. Is the manuscript presented in an intelligible fashion and written in standard English?

Reviewer #1: Yes

Reviewer #2: Yes

6. Review Comments to the Author

Reviewer #1: The authors have satisfactorily responded to this reviewers comments. I agree with reviewer 2 that a paragraph with implications for care and research moving forward is a necessary addition.

Reviewer #2: The authors have addressed all my comments. I commend them for their efforts. I would like to see another survey in the future with additional questions that will help us improve patient care during a potential second wave of the pandemic.

7. PLOS authors have the option to publish the peer review history of their article (what does this mean?). If published, this will include your full peer review and any attached files.

Reviewer #1: No

Reviewer #2: No

---

## [Editor Report · Acceptance letter]

1 Oct 2020

PONE-D-20-17999R1 

Insights from a global snapshot of the change in elective colorectal practice due to the COVID-19 pandemic 

Dear Dr. Mason:

I'm pleased to inform you that your manuscript has been deemed suitable for publication in PLOS ONE. Congratulations! Your manuscript is now with our production department. 

Kind regards, 

on behalf of

Dr. Zhi Ven Fong 

Academic Editor

PLOS ONE